# Comparative Genomics and Phylogenetic Analysis of the Chloroplast Genomes in Three Medicinal *Salvia* Species for Bioexploration

**DOI:** 10.3390/ijms232012080

**Published:** 2022-10-11

**Authors:** Qing Du, Heyu Yang, Jing Zeng, Zhuoer Chen, Junchen Zhou, Sihui Sun, Bin Wang, Chang Liu

**Affiliations:** 1Institute of Medicinal Plant Development, Chinese Academy of Medical Sciences, Peking Union Medical College, Beijing 100193, China; 2Key Laboratory of Medicinal Plant Resources of Qinghai-Tibetan Plateau in Qinghai Province, College of Pharmacy, Qinghai Minzu University, Xining 810007, China; 3College of Pharmacy, Xiangnan University, Chenzhou 423000, China

**Keywords:** *Salvia bowleyana*, *Salvia splendens*, *Salvia officinalis*, chloroplast genome, comparative genomics, repeat analysis, hypervariable regions, DNA barcode, phylogenetic analysis

## Abstract

To systematically determine their phylogenetic relationships and develop molecular markers for species discrimination of *Salvia bowleyana*, *S. splendens*, and *S. officinalis*, we sequenced their chloroplast genomes using the Illumina Hiseq 2500 platform. The chloroplast genomes length of *S. bowleyana*, *S. splendens*, and *S. officinalis* were 151,387 bp, 150,604 bp, and 151,163 bp, respectively. The six genes *ndh*B, *rpl*2, *rpl*23, *rps*7, *rps*12, and *ycf*2 were present in the IR regions. The chloroplast genomes of *S. bowleyana*, *S. splendens*, and *S. officinalis* contain 29 tandem repeats; 35, 29, 24 simple-sequence repeats, and 47, 49, 40 interspersed repeats, respectively. The three specific intergenic sequences (IGS) of *rps*16-*trn*Q-UUG, *trn*L-UAA-*trn*F-GAA, and *trn*M-CAU-*atp*E were found to discriminate the 23 *Salvia* species. A total of 91 intergenic spacer sequences were identified through genetic distance analysis. The two specific IGS regions (*trn*G-GCC*-trn*M-CAU and *ycf*3-*trn*S-GGA) have the highest K2p value identified in the three studied *Salvia* species. Furthermore, the phylogenetic tree showed that the 23 *Salvia* species formed a monophyletic group. Two pairs of genus-specific DNA barcode primers were found. The results will provide a solid foundation to understand the phylogenetic classification of the three *Salvia* species. Moreover, the specific intergenic regions can provide the probability to discriminate the *Salvia* species between the phenotype and the distinction of gene fragments.

## 1. Introduction

The Lamiaceae family is the sixth-largest family of flowering plants. It includes 10 subfamilies, 220 genera, and 3500 species [1]. Most of the species are mainly distributed in Asia, Europe, and Africa. In historical evolution, the family of Lamiaceae is most closely related to the family of Verbenaceae and Violinaceae [2]. In China, 99 genera and more than 800 species in the Lamiaceae family are found, which include about 1050 *Salvia* species. Among them, 78 varied species and 32 variants mostly grow in tropical or temperate areas [3]. Regarding the classification development of the *Salvia* genus, Bentham [4] once divided it into four subgenera and 12 groups and Briquet [5] divided it into 8 subgenera and 17 groups. There are also different taxonomic studies about the *Salvia* genus in the different regions, for example, subgenus *Calosphace* was divided into 91 groups by the American scientist Carl Epling [6], and the genus increased to 102 within the subsequent 20 years. In Europe and Africa, the *Salvia* genus is divided into four subgenera and eight groups in the flora of the USSR [7], whereas in the flora of Europe, the *Salvia* genus is divided into five groups [8]. The botanist of academician Wu Zhengyi in East Asia [9] divided the Chinese *Salvia* genus int o five subgenus groups and 18 subbranches. Based on the molecular systems and data of *rbc*L and *trn*L-F, the genus *Salvia* is not monophyletic; it has the relationship of sister taxa embedded with the genera of *Rosmarinus*, *Perovskia*, *Dorystaechas*, *Meriandra*, and *Zhumeria* [10]. Meanwhile, through molecular systematics and morphological evidence, 15 species from the 5 genera of *Rosmarinus*, *Perovskia*, *Spear*, *Meriandra*, and *Zhumeria* were formally merged into the generalized *Salvia* genus with 10 identified independent clades [11]. The 11 *Salvia* species in Japan were clustered in one branch based on the comparative data of *rbc*L, *trn*L-F, and ITS sequences [12]. The molecular systems of 38 *Salvia* species in China were classified using the ITS, *rbc*L, *psb*A-*trn*H, and *mat*K sequences in China, showing that the *Salvia* genus was clustered into one clade from China and Japan, except for the species of *Salvia* deserta, and the three subgenera defined in Chinese plants are not the monophyletic groups [13]. Using a study on the divergence of ITS, ETS, *psb*A-*trn*H, *ycf*1-*rps*15, *trn*L-*trn*F, and *rbc*L sequences, the phylogenetic tree containing 78 species and 10 variants confirmed that the *Salvia* of East Asian is a monophyletic group, formally naming the clade IV (S. Glutinaria Clade) as East Asian *Salvia* with eight groups [14]. More interestingly and meaningfully, the 345 species belonging to 77 Lamiaceae genera have been classified and clustered into phylogenetic groups based on the aspects of phytochemical constituents and treatment of the various disorders through the analysis of NRI and NTI metrics. The results showed that the *Salvia boweyara* had an effect on the treatment of reproductive and hepatic disorders [15]. Therefore, there are certain differences in the *Salvia* species from the aspects of their morphological characters, chemical composition, treatment effects on diseases, and molecular markers. We are looking forward to carrying out the integration of taxonomic research from various aspects to elucidate the classification status of the *Salvia* genus in the family.

The chloroplast is the essential organelle in plants. The chloroplast genome contains a variety of genes closely related to photosynthesis [16], evolution [17], and applications in genetic engineering [18]. In general, the chloroplast genome encodes more than 120 genes. These genes can be divided into three types [19] related to transcription and translation, photosynthesis, and the biosynthesis of amino acids and fatty acids. The genes distributed in the large-single copy (LSC) and small-single copy (SSC) regions are mainly related to photosynthetic systems I (PSA) and systems II (PSB). They also include large subunits of Rubisco (encoded by rbcL) [20], the *tRNA* gene (tRNA), the ATP enzyme gene (ATP), the NADH plastid-masking oxidoreductase gene (NADH), and the RNA polymerase gene (RPO) [21]. The genes distributed in the IRs region are mainly the genes encoding rRNA (RPS), including 16S and 23S genes, the intermediate genes being separated by encoding 4.5S rRNA, and 5S rRNA and 2tRNA genes, and some genes with unknown gene function [22].

The genes from chloroplast genomes can be used in species identification [23], phylogenetic evolution [24], genetic transformation [25], and molecular breeding of medicinal plants [26], providing basic data for resource identification and conservation. The sequences in the chloroplast genomes of medicinal plants, such as *psb*A-*trn*H, *mat*K, and *rbc*L, have been widely used for DNA molecular identification, and have now been developed for the analysis of polymorphic locus combinations of multiple genes and gene spacers [27]. To date, the chloroplast genomes of the 14 *Salvia* species in the Lamiaceae family have been reported [28,29,30].

Compared with the diversification of nuclear and mitochondrial genomes, the comprehensive development of chloroplast genomes could provide a basic database for further exploration regarding structural variation, characteristics, genetic evolution, and chemicals. Therefore, we sequenced and analyzed the chloroplast genomes of three *Salvia* species for the first time to identify divergence hotspots of phylogenetic genome regions and detect the applicability of phylogenomics for further resolving the evolutionary and systematic relationship in the *Salvia* genus of the Lamiaceae family.

## 2. Results

### 2.1. Morphological Characteristics of the Three Salvia Species

The three *Salvia* species have the common specifications of the Lamiaceae family: quadrangular stem, opposite leaves, corolla flower lip, and four nutlets. However, they have the obvious distinction from the phenotype of flower colors (1) varying from pink and purple (*S. bowleyana* and *S. officinalis*) to red (*S. splendens*). Moreover, the three *Salvia* species are perennial herbs with oblong or oval leaves (2), cymose inflorescences (3), and nutlets. Nevertheless, for *S. bowleyana*, the leaves are glabrous on both sides, only the veins are slightly pilose, and the top of the fruit is hairy (Figure 1a). For *S. splendens*, the stems, leaves on both sides, and petioles are not glabrous with glandular spots below. The fruits have irregular folds at the top, and narrow wings at the edge (Figure 1b). For *Salvia officinalis*, the stems, many branches, leaf surfaces, and petioles are covered with white short villi. The fruits are smooth and hairless (4) (Figure 1c) [1].

### 2.2. Gene Compositions Comparison of 23 Salvia Species

Schematic representations of *S. bowleyana*, *S. splendens*, and *S. officinalis* chloroplast genomes are shown in Figure 2, respectively. The total assembled length of them was 151,387 bp, 150,604 bp, and 151,163 bp, respectively. The lengths of LSC, SSC, and dual inverted repeat (IR) regions in the three chloroplast genomes were 82,772 bp, 17,573 bp, and 51,042 bp for *S. bowleyana*; 82,181 bp, 17,857 bp, and 50,566 bp for *S. splendens*; 82,429 bp, 17,510 bp, and 51,224 bp for *S. officinalis*. The GC contents of the three chloroplast genomes were 38.01%, 38.04%, and 38.04%, respectively (Table 1, Appendix A).

The chloroplast genomes of *S. bowleyana*, *S. splendens*, and *S. officinalis* contained 131, 130, and 131 genes, respectively, including 80, 79, and 80 protein-coding genes, 36 tRNA genes, and 8 rRNA genes (Appendix A). There are 14 PCGs (*rps*12 (×2), *rps*7 (×2), *rpl*2 (×2), *rpl*23 (×2), *ndh*B (×2), *ycf*2 (×2), and *ycf1*5 (×2)), 14 tRNA genes (*trn*A-UGC (×2), *trn*E-UUC (×2), *trn*M-CAU (×2), *trn*L-CAA (×2), *trn*N-GUU (×2), *trn*R-ACG (×2), *trn*V-GAC (×2)), and 8 rRNA genes (*rrn*16S (×2), *rrn*23S (×2), *rrn*4.5S (×2), and *rrn*5S (×2)) located in the both IRa and IRb regions (Table 1), respectively. Among the three genomes, twenty-two genes commonly exhibited introns, of which seven tRNA genes (*trn*K-UUU, *trn*L-UAA, *trn*C-ACA, *trn*E-UUC (×2), and *trn*A-UGC (×2)), and twelve cis-splicing CDS genes (*rps*16, *atp*F, *rpo*C1, *ycf*3, *clp*P, *pet*D, *rpl*16, *rpl*2 (×2), *ndh*B (×2), and *ndh*A) had a single intron. In particular, the three genes had one intron in the special species, of which both genes *trn*T-CGU and *pet*B are identified in the species of *S. bowleyana* and *S. splendens*. IN contrast, the protein-coding gene *pet*B was only shown in *S. officinalis.* Notably, two CDS genes of *ycf*3 and *clp*P displayed two introns and three exons (Table 2, Appendix A). Additionally, those containing the intron gene *trn*K-UUU, making up the *mat*K, had the largest intron in the three chloroplast genomes of *Salvia* species (2522 bp, 2494 bp, and 2517 bp, respectively). Except for the plants of Pteridophyta and parasitic species, the chloroplasts of land plants commonly contain the *mat*K mature enzyme gene in the intron of the lysine tRNA-K (UUU) gene, for instance, species of *Cuscuta* genus [31,32,33], which acts as a splicing factor for introns of the highly structured ribozyme group II [34,35]. Furthermore, the three segments of *rps*12 genes were located in the region of LSC, IRa, and IRb of the chloroplast genomes, respectively. The *rps*12 gene was split into two introns; one intron between exon 2 and 3 was 528 bp in length, and another intron between exon 1 and 2 was about 28 kb in length (Table 2, Appendix A). The latter intron is trans-spliced to produce mature *rps*12 mRNA (Appendix A) [36]. The exon 1 and the two copies of exons are trans-spliced together to form two transcripts. The arrows indicate the sense direction of the genes (Appendix A).

Among the 23 *Salvia* species, the lengths of the total genome, LSC, SSC, and IR varied from 150,604 bp to 153,995 bp, from 82,129 bp to 84,775 bp, from 17,464 bp to 17,875 bp, and from 25,283 bp to 25,815 bp, respectively. The percentage of GC contents for the total genome, LSC, SSC, and IRs regions varied from 37.94% to 38.05%, from 36.07% to 36.23%, from 31.63% to 32.07%, and from 43.06% to 43.20%. The gene numbers of the total genes, protein-encoding genes, and tRNA genes ranged from 130 to 133, from 85 to 88, and from 36 to 37, respectively. The chloroplast genomes in all 23 *Salvia* species encoded two copies of rrn16S, rrn23S, rrn4.5S, and rrn5S (Appendix A).

### 2.3. Gene Loss Analysis of the Chloroplast Genomes from 41 Species in the Lamiaceae Family

The gene losses of chloroplast genomes were analyzed in the 41 species of the Lamiaceae family that originated from the phylogenetic tree (Table 3). These species originated from 8 genera (*Salvia*, *Rosmarinus*, *Agastache*, *Dracocephalum*, *Ajuga*, *Leonurus*, *Elsholtzia*, and *Caryopteris*) of the Lamiaceae family. In the dual IR regions of chloroplast genomes, one of the *rpl*20 genes was stable and found in all 41 species; however, another one was found only in *D. heterophyllum*. Therefore, the intact *rpl*20 gene often can be used as the molecular signature gene in the angiosperm [37]. In addition, one of the *ycf*1 genes was across the SSC and IRb regions, the other pseudogene was across the SSC and IRa regions. Loss of the first *ycf*1 gene was observed in five chloroplast genomes of *A. campylanthoides*, *A. ciliata*, *A. decumbens*, *A. lupulina*, and *A. nipponensis*. Loss of the second one was not found in the chloroplast genomes of twenty-eight species except in the twelve chloroplast genomes from the six *Salvia* genus (*S. digitaloides*, *S. daiguii*, *S. meiliensis*, *S. chanryoenica*, *S. yangii*, and *S. nilotica*), *A. rugosa*, the four *Dracocephalum* genus (*D. heterophyllum*, *D. taliense*, *D. tanguticum*, and *D. moldavica*), and *L. japonicas*. As reported, in a total of 420 species, 357 species could be distinguished using *ycf*1 by means of specific primers designed for the amplification of these regions [38]. Moreover, the losses of the *ycf*15 genes occurred in five chloroplast genomes (*S. hispanica*, *S. tiliifolia*, *S. chanryoenica*, *A. forrestii*, and *E. densa*). Although the gene function of *ycf*15 genes is unknown, the transcriptome analyses of the *Camellia* genus revealed that the *ycf*15 gene was transcribed as a precursor polycistronic transcript which contained *ycf*2, *ycf*15, and antisense *trn*L-CAA [39]. Furthermore, the six genes in the LSC region, e.g., *pet*N, *acc*D, *rps*2, *rps*16, *rps*18, and *rps*19 were absent in the chloroplast genomes of *C. trichosphaera*, *R. officinalis*, *D. moldavica*, *E. densa*, *D. heterophyllum*, and *L. japonicus*, respectively. In contrast, in the SSC region, loss of the *rpl*32 and *ndh*D genes was found only in *S. splendens* and *C. mongholica* chloroplast genomes, respectively. Surprisingly, loss of the *rpl*32 gene can be transferred to the nucleus from the chloroplast genome of *Euphorbia schimperi* and this can be verified through the method of being sequenced in the nuclear transcriptome of *E. schimperi* (Table 3) [40]. The type of gene loss was mostly affirmed to be consistent with the topology of the evolutionary tree.

### 2.4. Analysis of Simple Sequence Repeats Polymorphism in the 23 Salvia Chloroplast Genomes

Repeat sequences have been commonly used as genetic markers to understand the evolution of the genus in the same family. Scattered (interspersed) repetition and tandem repetition sequences consisting of simple sequence repeats (SSRs) were analyzed in the 23 *Salvia* chloroplast genomes (Appendix A, Figure 3). We analyzed the content and percentage of SSR sequences in the 23 *Salvia* species. The results showed that 16, 12, and 10 SSR contained ″A″ as the repeat unit and 18, 14, and 14 SSR contained ″T″ as the repeat unit among the total 34, 26, and 24 mononucleotide repeats (Appendix A) in the chloroplast genomes of *S. bowleyana*, *S. splendens*, and *S. officinalis*, respectively. Moreover, the mononucleotide numbers of ″A″ and ″T″ as the repeat unit have an obvious difference. From the statistical results, the number of Poly A and Poly T repeats varied from 6 (*S. yangii*) to 16 (*S. bowleyana* and *S. miltiorrhiza f. alba*), from 9 (*S. plebeia*) to 21 (*S. prattii*). Rare numbers of Poly C and Poly G repeats were found only in the chloroplast genomes of *S. hispanica*, *S. plebeia*, and *S. meiliensis* [41]. One SSR with ″AT″ as the repeat unit was found in the eight *Salvia* chloroplast genomes of *S. splendens*, *S. digitaloides*, *S. daiguii*, *S. hispanica*, *S. tiliifolia*, *S. chanryoenica*, *S. prattii*, and *S. roborowskii*. Di-nucleotide SSR contained ″TA″ as the repeat unit in twelve chloroplast genomes of *S. bowleyana*, *S. bulleyana*, *S. przewalskii*, *S. yunnanensis*, *S. miltiorrhiza f. alba*, *S. chanryoenica*, *S. prattii*, *S. roborowskii*, *S. splendens*, *S. daiguii*, *S. hispanica*, and *S. tiliifolia*, respectively. Nevertheless, one trinucleotide SSR with ″AAT″ as the repeat unit was found in the chloroplast genome of *S. yunnanensis*
*(*Appendix A). The mononucleotide repeat unit is the most abundant type of the SSR repeats and it accounted for the proportion from 88% to 100% through comprehensive statistics of chloroplast genomes in the 23 *Salvia* species.

### 2.5. Repeat Sequences Analysis in the Chloroplast Genomes of 23 salvia Species

Except for in the SSR analysis of the 23 *Salvia* chloroplast genome, 29 tandem repeats by each species were identified for all the four kinds of tandem repeats, including the forward repeats, reverse repeats, palindromic repeats, and complement repeats in the chloroplast genomes of *S. bowleyana* (11 forward repeats, 3 reverse repeats, and 15 palindromic repeats), *S. splendens* (11 forward repeats, 4 reverse repeats, and 14 palindromic repeats) and *S. officinalis* (10 forward repeats, 5 reverse repeats, and 14 palindromic repeats), respectively. The greatest numbers of repeat types were forward repeats and palindromic repeats, while the numbers of reverse repeats and complement repeats were less and the latter were found only in the six chloroplast genomes, including *S. przewalskii*, *S. daiguii*, *S. meiliensis*, *S. merjamie*, *S. yangii*, and *S. nilotica*. The comparison of the number of predicted tandem repeats is shown in Appendix A, and Figure 3c.

Among the 23 *Salvia* chloroplast genomes of the interspersed repeats, the number of palindromic and direct repeats varied from 14 (*S. merjamie*, *S. sclarea*, and *S. daiguii*) to 26 (*S. miltiorrhiza*, *S. petrophila*, *S. prattii*, *S. roborowskii*, and *S. splendens*). The number of tandem repeats will be reduced by more than half and diversified from 6 (*S. bowleyana*, *S. splendens*, *S. plebeia*, *S. miltiorrhiza*, and *S. miltiorrhiza* f. *alba*) to 24 (*S. japonica*) while the similarity among the repeat unit sequences ≥ 90%. The e-values of interspersed repeats varied from 7.65 × 10^−23^ to 6.07 × 10^−^^4^. In this study, 47 interspersed repeats (25 palindromic repeats and 22 direct repeats), 49 interspersed repeats (23 palindromic repeats and 26 direct repeats), and 40 interspersed repeats (20 palindromic repeats and 20 direct repeats) were identified in the chloroplast genomes of *S. bowleyana*, *S. splendens*, and *S. officinalis*, respectively, with the length of repeat units 1, 2 being between 30 bp and 63 bp (Figure 3, Appendix A).

### 2.6. Structures of the IR Boundaries and Gene Features from 23 Salvia Species

The IR boundaries′ structure was analyzed in the 23 *Salvia* chloroplast genomes of the Lamiaceae family. From the analysis, six distinct genes, *rpl*22, *rps*19, *rpl*2 (×2), *ycf*1, *ndh*F, and *psb*A, were most explicitly found in the diverse regions or at the border regions of 23 chloroplast genomes (Figure 4). Furthermore, the variation range of these gene lengths was similar and did not exceed 2%. The genes of *rpl*22 and *psb*A were located in the LSC region, whereas *rpl*2 genes were located in the two IR regions in these species. One of the *rps*19 genes was located at the border area of LSC and IRb in all species. In addition, small fragments of the *rps*19 genes (*rps*19 pseudogene) were found at the border regions of the LSC and IRa in the fourteen chloroplast genomes of *S. bulleyana*, *S. digitaloides*, *S. japonica*, *S. plebeia*, *S. przewalskii*, *S. miltiorrhiza*, *S. daiguii*, *S. miltiorrhiza f.alba*, *S. meiliensis*, *S. petrophila*, *S. yangii*, *S. nilotica*, *S. prattii*, *S. roborowskii*. In contrast, the *ycf*1 genes traversed the border regions of SSC and IRb in all 23 *Salvia* species, while *ycf*1 gene fragments (*ycf*1 pseudogene) were found at the border regions of SSC and IRa in six *Salvia* chloroplast genomes (*S. merjamie*, *S. digitaloides*, *S. daiguii*, *S. chanryoenica*, *S. nilotica*, and *S.yangii*). Besides, *ndh*F genes were located at the border regions of IRa and SSC in all 23 species. The IRa/LSC boundary positions were located on the *trn*H genes in the five chloroplast genomes of *S. chanryoenica*, *S.splendens*, *S. nilotica*, *S. yangii*, and *S. tiliifolia.* Notably, a fragment of the *trn*N gene located in the IRb region of the *Salvia splendens* chloroplast genome (Figure 4) is often found in the *Cymbidium* genus among the photosynthetic orchids [42].

### 2.7. The Discrepancy of the 23 Salvia Chloroplast Genomes

The structures of chloroplast genomes are highly conserved. The medicinal plants can be accurately identified and distinguished by the comparison of barcodes from the whole chloroplast genome. The sequences of chloroplast genomes in the 23 *Salvia* species were analyzed using mVISTA, and the alignments were visualized with the *Salvia bowleyana* chloroplast genome as the reference genome (Appendix A). We found the sequences of 23 *Salvia* chloroplast genomes were mostly identically conserved except for the three variable areas located in the intergenic regions of the LSC region. The first one was the IGS region (*rps*16-*trn*Q-UUG) found in the nine *Salvia* chloroplast genomes (*S. officinalis*, *S. japonica*, *S. sclarea*, *S. meiliensis*, *S. hispanica*, *S. tiliifolia*, *S. yangii*, *S. splendens*, *S. nilotica*) (Appendix A (A)). The second one was the IGS region (*trn*L-UAA-*trn*F-GAA) varied in the chloroplast genome of *S. chanryoenica* (Appendix A (B)). The last one was the IGS region (*trn*M (cau)-*atp*E) diversified in the three chloroplast genomes of *S. chanryoenica*, *S. hispanica*, and *S. japonica* (Appendix A (C)).

### 2.8. Identification and Cloning of Hypervariable Regions

It is significant to develop molecular markers in the chloroplast genomes of plants by identifying the highly variable sites. In general, the large K2p distances indicate a high degree of sequence divergences. We analyzed the genetic distance among the IGS regions in the chloroplast genomes of 23 *Salvia* species. The results showed that K2p distances of 91 IGS regions ranged from 0.00 to 21.03 (Appendix A). Among them, 30 IGS regions had K2p distances varying from 3.52 to 21.03 (Figure 5a). Particularly, five IGS regions had higher K2p values diversified from 5.80 to 21.03, which were the regions of *trn*L-UAG-*ccs*A (21.03), *rps*16-*trn*Q-UUG (13.19), *ccs*A-*ndh*D (7.68), *rps*15-*ycf*1 (6.40), and *ndh*E-*ndh*G (5.80). Thus, these five regions of IGS can be suitable candidates for developing molecular markers in the 23 *Salvia* species. Meanwhile, the five IGS regions with higher K2p values were identified in the three studied *Salvia* species including *rps*16-*trn*Q-UUG (21.35), *trn*G-GCC-*trn*M-CAU (12.91), *ccs*A-*ndh*D (12.14), *ycf*3-*trn*S-GGA (10.92), and *rps*15-*ycf*1 (9.67) (Figure 5b, Appendix A).

Interestingly, the two IGS regions of *trn*G-GCC-*trn*M-CAU (Appendix A, M1) and *ycf*3-*trn*S-GGA (Appendix A, M2) were specific in the three studied species. We cloned the two regions and acquired the sequences of M1 (~300 bp) and M2 (~800bp) using Sanger sequencing (Appendix A). Then, we comparatively analyzed the two molecular markers (MMs) among the three studied *Salvia* species to determine the variations, including indels and single nucleotide polymorphisms (SNP) (Table 4, M1 and M2). The amplification products of the two IGS were checked and the strips were clearly shown on the agarose gel (Appendix A). From the peak map (up) and sequencing results (down) of the three studied *Salvia* species with the pairs of primers from M1 and M2 (Figure 6), four variant loci of SNP or indels were found among them and marked A, B, C, and D, respectively, at Figure 6. Therefore, the three *Salvia* species can be successfully discriminated based on these SNP and indel loci by separately or unitedly using the two M1 and M2 molecular markers. The intergenic region′s SNP (iSNP) has the potential to directly affect the protein structures or expression levels in accordance with the particular localization; therefore, it may affect the plant traits or genetic mechanisms [43]. In contrast to markers of the *Salvia* genus, two markers derived from the IGS regions of *pet*N-*psb*M and *psa*J-*rpl*33 can be successfully used to distinguish the five *Alpinia* species [44].

### 2.9. Identification and Comparison of the Genus-Specific DNA Barcodes Primer and Sequences

Primers can be designed from highly variable intergenic spacer sequences for PCR amplification. Then, we can distinguish the 23 *Salvia* species in the Lamiaceae family by sequence alignment and analysis using ecoPrimers software. After comparison, the two conservative intervals can be amplified through the designed PCR amplification primers to distinguish the 23 *salvia* genus. The primer sequences are shown in Table 4 (M3 and M4). Surprisingly, the two pairs of primers can be used to amplify the sequences of *trn*M-CAU-*atp*E and *ccs*A-*ndh*D after comparison between the *Salvia* chloroplast genomes and the BlastN database. Furthermore, the alignment results based on the blast database indicate that the two pair primers can also especially suit other distinct species, e.g., *Scutellaria* genus (Lamiaceae), *Camellia* genus (Theaceae), *Styrax* genus (Styracaceae), *Melissa* genus (Lamiaceae), *Eucalyptus* genus (Myrtaceae), etc.

### 2.10. Phylogenetic Analysis

The sequences of chloroplast genomes are a valuable database for the research of the evolutionary relationship in plants. To determine the phylogenetic positions of the three *Salvia* species in the Lamiaceae family, 80 protein sequences were extracted using the PhyloSuite software from the 43 chloroplast genomes in the species (Appendix A). Among them, 25 shared CDS proteins sequences were found present in 43 species, including *rpl*14, *rpl*33, *rpl*36, *rps*7, *rps*14, *psb*B, *psb*C, *psb*D, *psb*E, *psb*F, *psb*N, *psa*B, *psa*C, *psa*I, *pet*A, *pet*G, *pet*L, *ndh*C, *ndh*G, *cem*A, *atp*A, *atp*B, *atp*H, *atp*I, and *ycf*4 genes. We identified 29 proteins shared by 37 Lamiaceae species. However, there were only 25 proteins commonly shared in the studied 43 species. The other four proteins, including *atp*E, *psb*A, psbJ, and *psb*M, were only shared in 37 Lamiaceae species. The multiple sequence alignments of the 29 proteins are shown in Appendix A. Using *L. chuanxiong*
*(*Apiaceae family) and *P. notoginseng* (Araliaceae family) as the outgroups, the phylogenetic tree was generated by three methods of maximum likelihood (ML), maximum parsimony (MP), and neighbor-joining (NJ) based on the above-described data of whole chloroplast genomes. The three phylogenetic trees showed the same evolutionary relationship, in which 41 species including 37 species of the Lamiaceae family and four species of the Verbenaceae family were clustered together with 6 obvious clades. Among them, five species including *Dracocephalum* species (*D. heterophyllum*, *D. Taliense*, *D. tanguticum*, and *D. moldavica*) and *A. rugose* were clustered into one branch; in contrast, 23 *Salvia* species and one *Rosmarinus* species (*R. officinalis*) were clustered into one branch with six subbranches (Figure 7). In addition, six species from the *Ajuga* genus and four species from the *Caryopteris* genus were clustered into the other two branches, respectively. Single species of *L. japonicus* and *Elsholtzia densa* were gathered into one branch, partly, whereas the species of outgroups were more distantly related to other species. The ML bootstrap showed strong support with bootstrap values of 100% for eight nodes. The phylogenetic results resolved 26 nodes with bootstrap support values of 54–100 and that of 17 nodes were ≥ 74% (Figure 7).

## 3. Discussion

### 3.1. The Characteristics of Chloroplast Genomes and Genes in the Salvia Genus

In the chloroplast genomes of *S. bowleyana*, *S. splendens*, and *S. officinalis*, the total numbers of protein-coding genes were identical except that of *S. Splendens* was one less. The total numbers of tRNA and rRNA genes were the same as those of other *Salvia* species. These results indicated that the chloroplast genomes of the *Salvia* species were highly conserved. The selected 41 species from the Lamiaceae family and the two outgroup species (*L. chuanxiong* and *P. notoginseng*) possessed similar pharmacological effects, such as promoting blood circulation for removing blood stasis, increasing coronary flow, improving microcirculation, protecting the heart, improving the body hypoxia resistance, and having anti-hepatitis, antitumor, and antiviral effects [45]. Chloroplasts play an irreplaceable role in the formation of chemicals and the development of phenotypes due to the genes from nuclear, and mitochondrial genomes. However, the variability of the nuclear genome was found to be higher than that of the chloroplast genome and mitochondrial genome, as reported from the average genetic distance among all the strains of CWR and cultivated rice [46]. Therefore, it is indispensable to analyze the genetic divergence in the chloroplast genomes of *Salvia* species.

### 3.2. The Divergence between IGS Regions of the Salvia Genus Compared to Other Plants

It makes sense that the DNA sequences of the hypervariable regions and comparison of chloroplast genomes in three IGS regions of *rps*16-*trn*Q-UUG, *trn*L-UAA-*trn*F-GAA, and *trn*M(cau)-*atp*E can be used to distinguish the ten *Salvia* species (*S. officinalis*, *S. japonica*, *S. sclarea*, *S. meiliensis*, *S. hispanica*, *S. tiliifolia*, *S. yangii*, *S. splendens*, *S. nilotica*, and *S. chanryoenica*). The first IGS region has been found in the species of *Zingiber* officinale and *Cofeeae* alliance [47,48]. The second one commonly occurs in the angiosperm [49]. The last one has diversified and some parts of the oldest mtDNAs of *trn*V(uac)-*trn*M(cau)-*atp*E-*atp*B-*rbc*L were transferred from cpDNA to mtDNA since they have a common ancestor in extant gymnosperms and angiosperms [50]. As reported, the phylogenetic relationships in the Eurystachys clade were reconstructed utilizing nuclear ribosomal DNA sequences (nrETS, 5S-NTS) from 148 accessions into 12 well-supported genera, including widely recognized and well-defined segregates such as *Prasium* and *Sideritis* [51].

In contrast, the special IGS regions of the two iSNPs, namely *trn*G-GCC-*trn*M-CAU and *ycf*3-*trn*S-GGA, were used to discriminate the three studied *Salvia* species. In previous studies, most of the SNPs were found in intergenic sequences, and the *trn*G-GCC-*trn*M-CAU was one of the maximum number of SNPs found four times to distinguish the six *Saccharum* species [52]. Meantime, the variable hotspot regions of *ycf*3-*trn*S-GGA also can be useful as the candidate DNA barcodes for Adoxaceae and Caprifoliaceae species, and also for assessing interspecific divergence in Dipsacales species [53]. In addition, research has shown that the *rps*14 gene can be used as a DNA barcode for the identification of 34 Lamiaceae species collected from plants in the Pakistan area [54].

Therefore, the DNA barcode primers identified in the study can be potentially developed for the identification and phytotaxonomy of genus *Salvia* species through the divergence IGS regions.

### 3.3. The Functional Features of IR Regions and Genes of the Salvia Genus together with Other Plants

The sequences of IR can complement a certain segment of the upstream sequence downstream of the same DNA strand. They can then form a hairpin structure with a double helix stem and a single-stranded ring with a DNA double helix. The sequence between two reverse repeat units forms a single chain loop. Two copies are separated by a sequence or no interval sequence, which is in reverse series, and will form a specific palindrome sequence (P) [55]. Compared to the IRLC between the Papilionoideae subfamily [56] and the Lamiaceae family, they have the four common genes of *ndh*B, *rpl*23, *ycf*1, and *ycf*15.

In the IR regions, the genes of *ndh*B, *rpl*2, *rpl*23, *rps*7, *rps*12, and *ycf*2 were present in the chloroplast genomes of 41 species, and these genes have a special function in the area of gene expressions. There are the five hypothetical coding regions genes of *ycf*1, *ycf*2, *ycf*4, *ycf*15, and two open reading frames (ORF42 and ORF56), which are also found in the chloroplast genomes of the other species, such as *Clerodendranthus spicatus* [57]. Both genes *ycf*3 and *ycf*4 were present in the LSC region of the 41 species′ chloroplast genomes. The sequence of *ycf*3 is conserved in plants and contains three tetratrico-peptide repeats (TPR), which can act as the functions essential for the accumulation of the photosystem I (PSI) complex through a post-translational level [58,59]. The *ycf*4 gene forms modules that mediate PSI assembly and facilitate the integration of peripheral PSI subunits and LHCIs into the PSI reaction center subcomplex [60].

## 4. Materials and Methods

### 4.1. Plant Photos and Materials

*Salvia bowleyana*, *S. splendens*, and *S. officinalis* are the three characteristic plants from the *Salvia* genus of the Lamiaceae family. The photos of *Salvia bowleyana* and *S. splendens* were provided by the Jiangsu Nanjing Botanical Garden and the Civic Park of Guangdong, and identified by Professor Peng LQ (Chuzhou Hospital of Integrated Traditional Chinese and Western medicine, Anhui Province). In addition, the *S. officinalis* photo is from Dr. Qi YD′s team (Dr. Zhao Xinlei, Institute of Medicinal Plant Development, Chinese Academy of Medical Sciences, Peking Union Medical College, Beijing, zhaoxinlei2009@sina.com) (Figure 1). Furthermore, we collected the young leaves of *S. bowleyana*, *S. splendens*, and *S. officinalis* from the Guangxi Medical Botanical Garden, Nanning, Guangxi, China (Geospatial coordinates: 22°51′35.9″ N, 108°23′00.5″ E) and dried them with silica gel immediately for total genomic DNA isolation and sequencing of the chloroplast genome. The voucher specimens were deposited at the Institute of Medicinal Plant Development under the voucher number: implad201910237, implad201808155, and implad20170492, respectively (contact person: HM Chen; email: hmchen@implad.ac.cn). Moreover, the fresh leaves of three plants were used to clone the DNA barcode sequences from Jiujiang city, Jiangxi province (29°11′36.6″ N, 114°47′52.9″ E), Songjiang, Shanghai city (30°56′49.5″ N, 121°15′23.3″ E), and Institute of Medicinal Plant Development, Chinese Academy of Medical Sciences, Peking Union Medical College, Beijing city (116°25′ E, 39°47′ N), and Shunyi Dist., Beijing city (116°46′56″ E, 40°5′41″ N).

### 4.2. DNA Extraction, Determination of DNA Quality, and PCR Amplification Products

Total genomic DNA was extracted from the 20 dried leaves for sequencing of chloroplast genome and fresh leaves were taken from the three single plants for cloning the DNA barcode sequences using a plant genomic DNA kit (Tiangen Biotech, Beijing, China). The extraction of DNA is a universal technology as the flowchart shows in Dr. Li′s research [61]. We firstly ground plant tissues with liquid nitrogen. Then, we added the GPS buffer, RNase A, GPA buffer, absolute ethyl alcohol, deprotein fluid RD, bleach solution PW, and elution buffer TB. Next, we loaded the collection solutions on a column. During the courses, we mixed the solution and centrifuged each step. Lastly, the DNA bound to the column was eluted with an elution buffer. The DNA purity and amplification products were detected by 1.0% agarose gel electrophoresis stained with ethidium bromide alongside a 100 bp ladder (New England Biolabs, Ipswitch, MA, USA) using the DNA marker as the reference to determine the size of the amplified fragments (Takara) [62]. Otherwise, DNA concentration was determined using the Nanodrop spectrophotometer 2000 (Thermo, Waltham, Massachusetts, USA). Furthermore, the extraction of chloroplast DNA (cpDNA) for whole plastid genome sequencing should undergo three stages: separation of chloroplasts from cells, purification of chloroplasts, and isolation of cpDNA [63].

### 4.3. Chloroplast Genome Sequencing, Assembly, Annotation, and Manual Curation

DNA extracts containing the DNA concentration of 500 ng were applied to construct a library with lengths of short-insert fragments of 500 bps. The library was sequenced in a pair-end model with a read length of 150 bp on an Illumina Hiseq 2500 platform in accordance with the MiSeq platform provided by the manufacturer′s directions [64]. The sequencing raw data were acquired from *S. bowleyana*, *S. splendens*, and *S. officinalis* with sizes of 7.1 Gbs, 6.8 Gbs, and 7.02 Gbs and 250bps pair-end read lengths, respectively. The raw data were submitted to the NCBI database and assigned the Sequence Read Archive (SRA) accession numbers SRR14415377, SRR17843445, and SRR17853381, respectively. The raw reads were filtered using Trimmomatic 0.35 with default parameters to remove adapters and low-quality bases [65]. The three chloroplast genomes were assembled using the NOVOPlasty (v 4.2) software [66] with the default parameters and the rbcL sequences as the seed. After that, we annotated these genomes using the CpGAVAS2 web service (http://www.herbalgenomics.org/cpgavas2/, accessed on 1 May 2022) [67]. The annotation errors were manually corrected using the Apollo software [68]. The assembly and the annotation results of *S. bowleyana*, *S. splendens*, and *S. officinalis* were submitted to GenBank with the accession numbers OM617845, OM617847, and OM617846, respectively.

### 4.4. Visualization and Analysis of Genome Content, cis- and Trans-Splicing genes

The chloroplast genome structure, cis-splicing genes, and trans-splicing PCGs were visualized using CPGview-RSG software (http://www.1kmpg.cn/cpgview/, accessed on 1 May 2022) [69]. The gene contents of 41 studied species (Appendix A) were analyzed including the length of the complete genome sequences and the four regions, all genes, CDS, tRNAs, and rRNAs.

### 4.5. Repeat Analysis

We annotated the repeat sequences using the CPGAVAS2 for the chloroplast genomes of *S. bowleyana*, *S. splendens*, and *S. officinalis*. The SSRs of 23 *Salvia* species were identified using MISA software (http://pgrc.ipk-gatersleben.de/misa/, accessed on 1 May 2022) [70], also called the microsatellite sequence. The minimum numbers of repeat units for mononucleotide, dinucleotide, trinucleotide, tetranucleotide, pentanucleotide, hexanucleotide, and hexagenucleotide were set as 10, 5, 4, 3, 3, 3 and 3, respectively. The minimum distance between the 2 SSRs was set to 100 bp. If the distance was less than 100 bp, the two SSRs were treated as a composite microsatellite. The tandem repeats sequence (TRS) of the 23 *Salvia* chloroplast genomes was predicted using the Tandem Repeats Finder (TRF) software [71]. The interspersed repeats sequence (IRS) was predicted using the REPuter program (https://bibiserv.cebitec.uni-bielefeld.de/reputer, accessed on 1 May 2022), with the parameters as follows: maximum computed repeats = 30 and minimal repeat size = 8) [72]. The comparison of the chloroplast genomes was conducted using VMATCH software (Professor Stefan Kurtz, Computer Science at the Center for Bioinformatics, University of Hamburg, Germany) [73].

### 4.6. Comparative Genomic Analysis

We downloaded 40 chloroplast genomes sequences from the GenBank database including 38 species from the Lamiaceae family and two outgroups (Ligusticum chuanxiong from the Apiaceae family and Panax notoginseng from the Araliaceae family, for further analysis. The boundaries of the LSC, SSC, and IR regions boundary of chloroplast genomes from 23 *Salvia* species were visualized using the IR scope software (https://irscope.shinyapps.io/irapp/) [74] and the characteristic genes including the diverse areas were analyzed. The chloroplast genome sequences of 23 species from *Salvia* genera were compared with the annotated *S. bowleyana* chloroplast as the reference using the mVISTA program in a Shuffle-LAGAN mode with default parameters (Rank VISTA probability threshold = 0.5) [75,76]. The genetic distances of IGS regions from the chloroplast genomes of 23 *Salvia* species were calculated using the distmat program from EMBOSS (v6.3.1) [77] with the Kimura 2-parameters (K2p) evolutionary model [78].

### 4.7. Primer Identification and Design, PCR Amplification, Sequencing, and Analysis of Genus-Specific DNA Barcode Sequences

To discover DNA barcode sequences that can distinguish the 23 *Salvia* species, especially the three studied species, we analyzed the PCR amplification primers from their chloroplast genome sequences using ecoPrimers software [79]. Moreover, the sequences of two pairs of primers were compared to the other species through the CBI Multiple Sequence Alignment Viewer (Version 1.21.0, Max Seq Difference = 0.75) from the BLASTN website (https://blast.ncbi.nlm.nih.gov/) [80]. The two pairs of specific primers were designed to differently amplify the specific IGS regions identified in the three studied *Salvia* species by the Primer 3 software [81]. The PCR amplification system for genus-specific DNA barcode sequences of each reaction included 12.5 µL of 2 Taq PCR Master Mix (TransGen Biotech), 1.0 µL of each primer (0.4 µM), 2.0 µL of extracted template DNA, and ddH_2_O added to a final volume of 25 µL [82]. A negative control (Milli-Q water in place of DNA template) was included in each PCR to ensure there was no contamination. All the amplifications were performed on a Pro-Flex PCR system (Applied Biosystems, Waltham, MA, USA) instrument with the amplification procedures: degeneration 94 °C for 2 min followed by 35 cycles of 94 °C for 30 s, 57 °C for 30 s, 72 °C for 60 s, and a final extension step at 72 °C for 2 min. The amplification products were saved at 4 °C and sequenced at SinoGenoMax Co., Ltd. using the Sanger sequencing platform with the same cloning primers on the ABI Prism 3730 Genetic Analyzer (Applied Biosystems, USA). The sequences were spliced and analyzed by the GeneDoc software (3.2) [83].

### 4.8. Phylogenetic Analysis

We developed phylogenetic analysis using the concatenated coding sequences (CDS) of the chloroplast genomes from 43 species. These include 37 Lamiales species (*S. bowleyana*, *S. splendens*, *S. officinalis*, *S. bulleyana*, *S. digitaloides*, *S. japonica*, *S. plebeia*, *S. przewalskii*, *S. yunnanensis*, *S. miltiorrhiza*, *S. daiguii*, *S. sclarea*, *S. meiliensis*, *S. miltiorrhiza f.alba*, *S. hispanica*, *S. merjamie*, *S. petrophila*, *S. tiliifolia*, *S. chanryoenica*, *S. yangii*, *S. prattii*, *S. roborowskii*, *S. nilotica*, *R. officinalis*, *A. rugosa*, *D. heterophyllum*, *D. taliense*, *D. tanguticum*, *D. moldavica*, *A. forrestii*, *A. campylanthoides*, *A. ciliata*, *A. decumbens*, *A. lupulina*, *A. nipponensis*, *L. japonicus*, and *Elsholtzia densa*) and 4 species of the Verbenaceae family (*C. trichosphaera*, *C. mongholica*, *C. incana*, and *C. forrestii*), while the two species *Ligusticum chuanxiong* from the Apiaceae family and *Panax notoginseng* from the Araliaceae family were used as the outgroup. The chloroplast genome sequences were downloaded from GenBank (Appendix A). The shared CDSs were extracted, concatenated using PhyloSuite (v1.2.2) [84], and aligned using MAFFT (v7.313) [85]. Moreover, the sequences of 29 CDSs with small variations among the 37 chloroplast genomes from the Lamiaceae family were compared using the Genedoc (3.2) [83]. Phylogenetic analysis was conducted based on three methods of maximum likelihood(ML), maximum parsimony (MP), and neighbor-joining (NJ) implemented in IQ-TREE (v1.6.8) [86] under the TVM+F+I+G4 nucleotide substitution model. The reliability of the phylogenetic tree was assessed by bootstrap analysis with 1000 replications and was visualized using MEGA-X [87].

## 5. Conclusions

The complete chloroplast genomes of *S. bowleyana*, *S. splendens*, and *S. officinalis* were acquired using Illumina sequencing technology. These three species can be easily discriminated from the phenotype. Phylogenetic analysis showed that 23 *Salvia* species and one *Rosmarinus* genus were clustered into one branch with six subbranches, of which the three studied species were included in the diverse branches. The sequence divergence found seven sites of IGS regions: *rps*16-*trn*Q-UUG, *trn*L-UAA-*trn*F-GAA, *trn*M-CAU-*atp*E, *trn*L-UAG-*ccs*A, *ccs*A-*ndh*D, *rps*15-*ycf*1, and *ndh*E-*ndh*G. Notably, the two IGS regions of *trn*G-GCC-*trn*M-CAU and *ycf*3-*trn*S-GGA were identified in the three studied *Salvia* species. The sequences′ divergence had a high variability and indicates they can be developed as DNA markers for further identification and phytotaxonomy of the *Salvia* genus. Overall, the data obtained will contribute to further development of the authentication, diversity, ecology, taxonomy, phylogenetic evolution, and conservation of the *Salvia* genus in China.

## Figures and Tables

**Figure 1 ijms-23-12080-f001:**
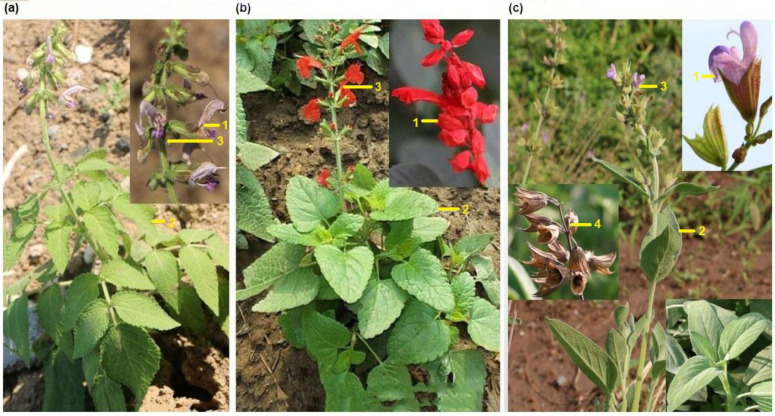
Three *Salvia* species of the Lamiaceae family. *S.*
*bowleyana* (**a**), *S.*
*splendens* (**b**), and *S.*
*officinalis* (**c**). The numbers 1–4 shown in yellow refer to the four different characteristics among the three species, which include the colors of flowers, shape of leaves, type of inflorescences, and appearance of fruits. 1: flower; 2: leaf; 3: inflorescences; 4: fruit.

**Figure 2 ijms-23-12080-f002:**
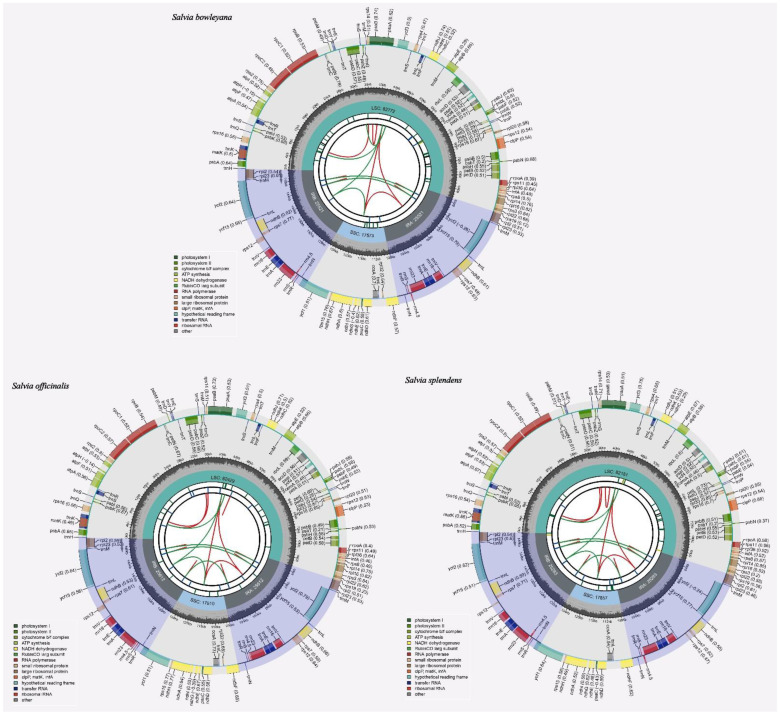
Graphic representation of features identified in the *S. bowleyana* (**a**), *S. splendens* (**b**), and *S. officinalis* (**c**) chloroplast genomes. Each map contains seven circles. From the center going outward, the first circle shows the distributed repeats connected with red (the forward direction) and green (the reverse direction) arcs. The next circle shows the tandem repeats marked with short bars. The third circle shows the microsatellite sequences as short bars. The fourth circle shows the size of the LSC and SSC. The fifth circle shows the IRA and IRB. The sixth circle shows the GC contents along the plastome. The seventh circle shows the genes having different colors based on their functional groups.

**Figure 3 ijms-23-12080-f003:**
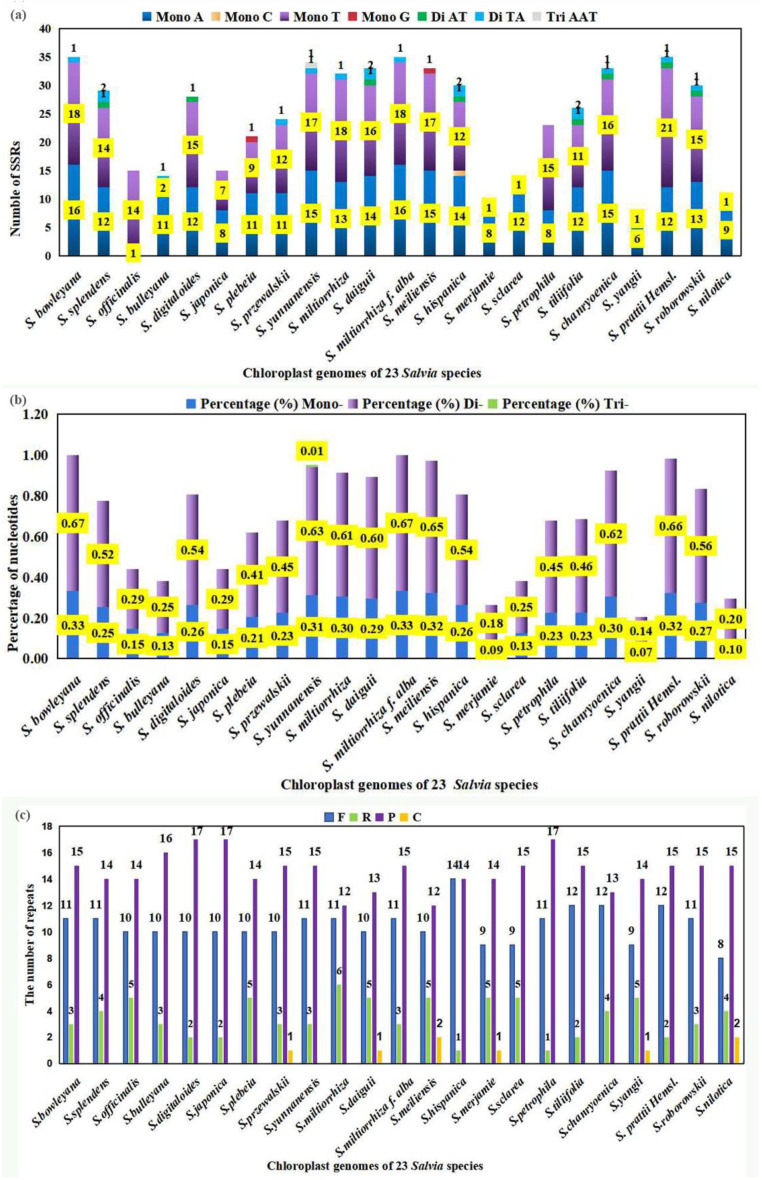
The repeats analysis in the 23 *Salvia* species. The number of diverse repeats has been marked on the strips in different colors. The abscissa represents the chloroplast genomes of 23 *Salvia* species; the ordinates represent the number of SSRs (**a**), the percentage of nucleotides (**b**), and the number of repeats (**c**). In (a), the different types of SSRs are filled in blue (mono A), orange (mono C), purple (mono T), red (mono G), green (di AT), blue (di TA), and gray (Tri AAT) together marked with the detailed quantum in yellow and black within the diverse columns. In (b), the percentage of mononucleotides, dinucleotides, and trinucleotides is filled in blue, purple, and green together marked with the detailed quantum in yellow and black within the diverse columns. In (c), the number of repeats in the types of forward repeats (F), reverse repeats (R), palindromic repeats (P), and complement repeats (C) is filled in blue, green, purple, and orange together marked with the detailed quantum in black above the diverse columns. Mono: mononucleotide; Di: dinucleotide; Tri: trinucleotide; F: forward repeats, R: reverse repeats; P: palindromic repeats, and C: complement repeats.

**Figure 4 ijms-23-12080-f004:**
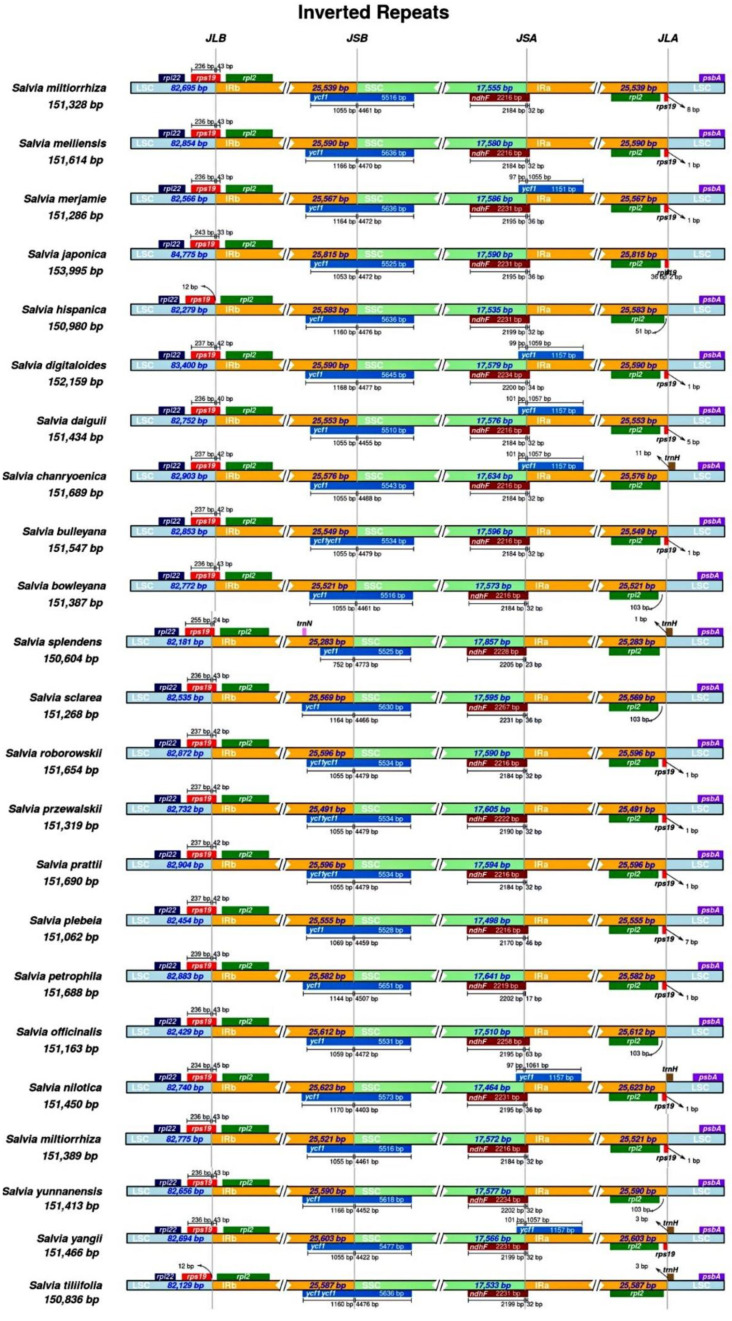
Comparison of the border areas among the LSC, SSC, and IR regions in the 23 *Salvia* chloroplast genomes. The genes are denoted by colored boxes. The gaps between the genes and the boundaries are indicated by the base lengths (bp). The thin lines represent the connection points of each area, and the information of the genes near the connection points is shown in the figures. The species′ Latin names and the length of the plastomes are shown on the left. The JLB, JSB, JSA, and JLA represent junction sites of LSC/IRb, IRb/SSC, SSC/IRa, and IRa/LSC, respectively. The distance from the start and end positions of different genes across junction sites is shown above or below the corresponding genes.

**Figure 5 ijms-23-12080-f005:**
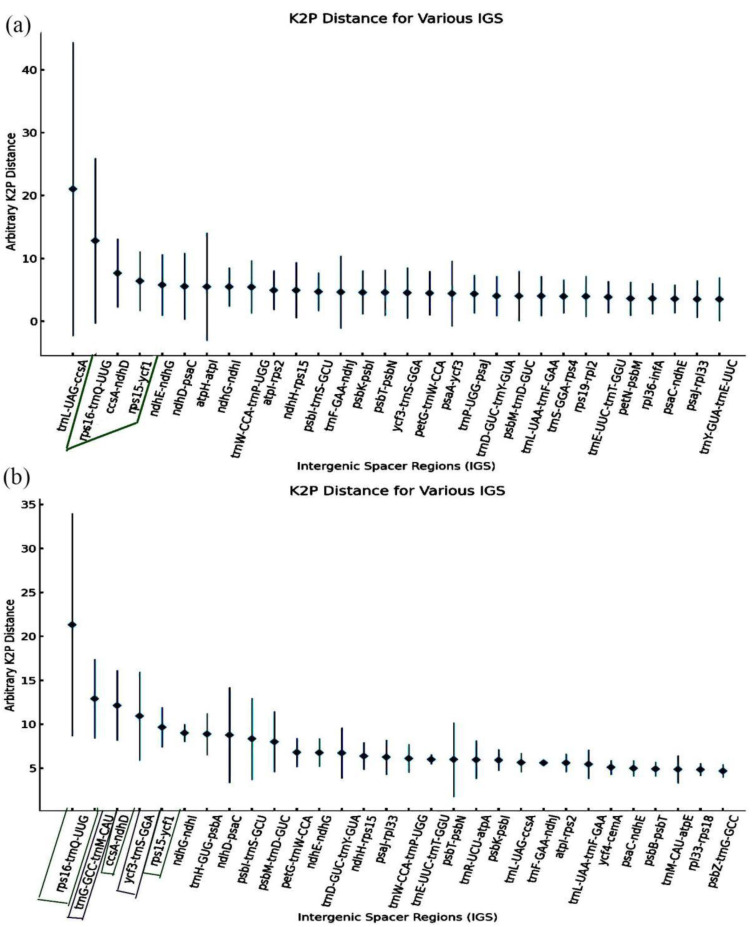
Average K2p distances for intergenic spacer regions in the chloroplast genomes of 23 *Salvia* species (**a**) and the three studies species (**b**) from the Lamiaceae family. The K2p distances were calculated among 23 *Salvia* chloroplast genomes in pairs. The black dots represent the average value of the three pairs. The error bars represent the standard error among the three pairs. Among the five IGSs with the highest K2p values, the IGSs marked in the green frame are common in the chloroplast genomes between 23 *Salvia* species and the three studies species, while the marked in purple are the specific IGSs in the chloroplast genomes of the three studies *Salvia* species.

**Figure 6 ijms-23-12080-f006:**
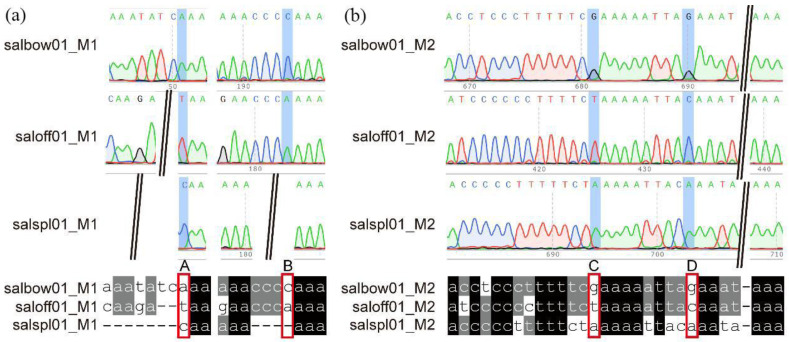
The peak map (up) and sequencing results (down) of the three studied *Salvia* species with the pairs of primers M1 (**a**) and M2 (**b**). The symbols of salbow01_M1 (**a**) and salbow01_M2 (**b**) are the sequencing results and peak map from one sample of *Salvia bowleyana;* the symbols of the saloff01_M1 and saloff01_M2 are the one sample of *Salvia officinalis*, and the symbols of salspl01_M1 and salspl01_M2 are the one sample of *Salvia splendens*. The variant bases have been marked A, B, C, and D in a red frame of the sequences.

**Figure 7 ijms-23-12080-f007:**
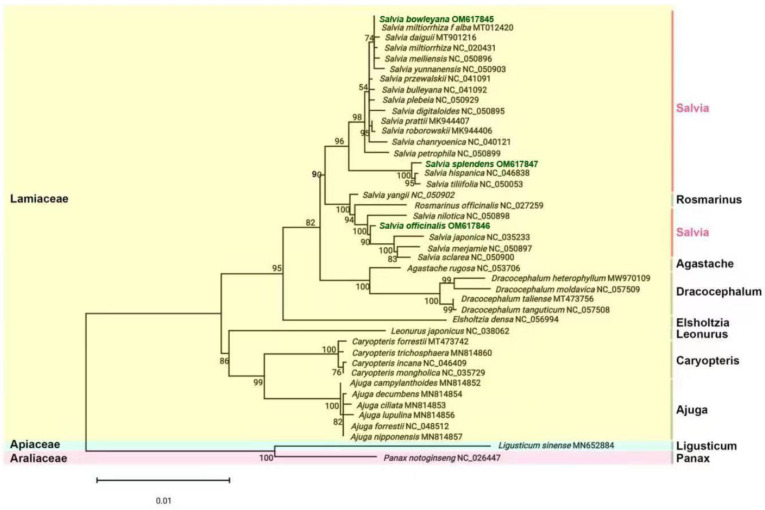
The phylogenetic relationships of the 43 species. These include 37 Lamiales species and 4 species of the Verbenaceae family, while the two species *Ligusticum chuanxiong* and *Panax notoginseng* were used as the outgroup. The tree was constructed with the sequences of 80 CDSs shared among all 43 species by using the three methods of maximum likelihood (ML), maximum parsimony (MP), and neighbor-joining (NJ). Bootstrap supports were calculated from 1000 replicates.

**Table 1 ijms-23-12080-t001:** Comparison of the gene contents in the chloroplast genomes of *Salvia bowleyana*, *Salvia splendens*, and *Salvia officinalis*.

Species/Items		*S. bowleyana*	*S. splendens*	*S. officinalis*
Gene Function	Gene Type	Gene Name		
tRNA	tRNA genes	36 *trn* genes (include one intron in 8 genes)	36 *trn* genes (include one intron in 8 genes)	36 *trn* genes (include one intron in 8 genes)
Photosynthesis	Subunits of ATP synthase	*atp*A, *atp*B, *atp*E, *atp*F, *atp*H, *atp*I		
	Subunits of photosystem Ⅰ	*psa*A, *psa*B, *psa*C, *psa*I, *psa*J		
	Subunits of photosystem Ⅱ	*psb*A, *psb*B, *psb*C, *psb*D, *psb*E, *psb*F, *psb*I, *psb*J, *psb*K, *psb*L, *psb*M, *psb*N, *psb*T, *psb*Z, *ycf*3		
Gene expression	Ribosomal RNAs	*rrn*16*s*^a^, *rrn*16*s*^b^, *rrn*23*s*^a^, *rrn*23*s*^b^, *rrn*4.5*s*^a^, *rrn*4.5*s*^b^, *rrn*5*s*^a^, *rrn*5*s*^b^		
	DNA-dependent RNA polymerase	*rpo*A, *rpoB*, *rpo*C1, *rpo*C2		
	Small subunit of ribosome	*rps*11, *rps*12^L^, *rps*12^a^, *rps*12^b^, *rps*14, *rps*15, *rps*16, *rps*18, *rps*19, *rps*2, *rps*3, *rps*4, *rps*7^a^, *rps*7^b^, *rps*8		
	Large subunit of ribosome	*rpl*14, *rpl*16, *rpl*2^a^, *rpl*2^b^, *rpl*20, *rpl*22, *rpl*23^a^, *rpl*23^b^, *rpl*32, *rp*l33, *rp*l36		
	Subunits of NADH-dehydrogenase	*ndh*A, *ndh*B^a^, *ndh*B^b^, *ndh*C, *ndh*D, *ndh*E, *ndh*F, *ndh*G, *ndh*H, *ndh*I, *ndh*J, *ndh*K		
	Subunits of cytochrome b/f complex	*pet*A, *pet*B, *pet*D, *pet*G, *pet*L, *pet*N		
	Ribulose diphosphate carboxylase subunit	*rbc*L		
Other genes	Subunit of acetyl-CoA-carboxylase	*acc*D		
	C-type cytochrome synthase	*ccs*A		
	Protease	*clp*P		
	Translation initiation factor	*inf*A		
	Mature enzyme	*mat*K		
	Envelope membrane protein	*cem*A		
Unknown functions	Conservative open reading frame	*ycf*1^s-b^, *ycf*2^a^, *ycf*2^b^, *ycf*15^a^, *ycf*15^b^, *ycf*4		

L: LSC region; a: IRa region; b: IRb region; s-b: Across the SSC and IRb regions.

**Table 2 ijms-23-12080-t002:** The lengths of introns and exons for the splitting genes in the chloroplast genomes of *S. bowleyana*, *S. splendens*, and *S. officinalis*.

Gene Name	Strand	Initial Position–Final Position			Length (bp)														
		*S. bowleyan*	*S. splendens*	*S. officinalis*	The First Exon			The First Intron			The Second Exon			The Second Intron			The Third Exon		
		A	B	C	A	B	C	A	B	C	A	B	C	A	B	C	A	B	C
*trn*K-UUU	-	1672–4266	1684–4250	1703–4292	37	37	37	2522	2494	2517	36	36	36						
*rps*16	-	4835–5945	4819–5917	4863–5972	40	40	40	874	862	873	197	197	197						
*trn*T-CGU	+	9001–9755	8765–9528	/	35	35	/	677	686	/	43	43	/						
*trn*S-CGA	+	/	/	8621–9377	/	/	32	/	/	665		/	60						
*atp*F	-	11,742–12,989	11,506–12,764	11,353–12,606	145	145	145	693	704	699	410	410	410						
*rpo*C1	-	20,712–23,525	20,528–23,339	20,399–23,215	430	430	430	759	757	762	1625	1625	1625						
*ycf*3	-	41,963–43,894	41,526–43,464	41,641–43,591	129	129	129	696	702	706	228	228	228	726	727	735	153	153	153
*trn*L-UAA	+	46,799–47,338	46,350–46,917	46,202–46,773	35	35	35	455	483	487	50	50	50						
*trn*C-ACA	-	50,870–51,518	50,236–50,881	50,440–51,087	38	38	38	555	552	554	56	56	56						
*rps*12^L^		68,691–68,804	68,105–68,218	68,355–68,468	114	114	114												
*clp*P	-	68,928–70,839	68,342–70,250	68,591–70,509	71	71	71	692	703	711	294	294	294	629	615	617	226	226	226
*pet*B	+	73,746–75,096	73,171–74,533	/	6	6	/	703	715	/	642	642	/						
*pet*D	-	75,290–76,492	74,721–75,904	74,979–76,169	8	8	8	720	701	708	475	475	475						
*rpl*16	-	79,937–81,217	79,325–80,600	79,599–80,867	9	9	9	873	868	861	399	399	399						
*rpl*2	-	82,875–84,357	82,266–83,757	82,532–84,019	391	391	391	658	667	663	434	434	434						
*ndh*B	+	93,058–95,211	92,464–94,617	92,711–94,918	721	721	775	675	675	675	758	758	758						
*rps*12^b^		96,061–96,844	96,018–96,260	95,714–96,507	114	114	114	/	/	/	232	243	232	528	/	538	26	/	26
*trn*E-UUC	+	100,535–101,546	99,979–100,997	100,210–101,229	32	32	32	940	947	948	40	40	40						
*trn*A-UGC	+	101,611–102,478	101,062–101,938	101,294–102,171	37	37	37	795	804	805	36	36	36						
*ndh*A	-	117,349–119,425	116,488–118,588	117,038–119,137	553	553	553	985	1009	1008	539	539	539						
*trn*A-UGC	-	131,682–132,549	130,848–131,724	131,422–132,299	37	37	37	795	804	805	36	36	36						
*trn*E-UUC	-	132,614–133,625	131,789–132,807	132,364–133,383	32	32	32	940	947	948	40	40	40						
*rps*12^a^		137,316–138,099	136,526–136,768	137,086–137,879	114	114	114	/	/	/	232	241	232	528	/	528	26	/	26
*ndh*B	+	138,949–141,102	138,169–140,322	138,675–140,882	721	721	775	675	675	675	758	758	758						
*rpl*2	+	149,803–151,285	149,029–150,520	149,574–151,061	391	391	391	658	667	663	434	434	434						

″+″ indicates a positive chain; ″-″ indicates a negative chain; A: *S. bowleyan*; B: *S. splendens*; C: *S. officinalis.* L: LSC region; a: IRa region; b: IRb region.

**Table 3 ijms-23-12080-t003:** Gene losses in the different regions of the 41 chloroplast genomes from the Lamiaceae family.

Genus	Name of Species	The Genes in the IR Region				The Genes in the LSC Region						The Genes in the SSC Region	
		*rpl*20_copy	*ycf*1	*ycf*1_*copy*	*ycf*15	*pet*N	*acc*D	*rps*2	*rps*16	*rps*18	*rps*19 *	*rpl*32	*ndh*D
*Salvia*	*S. bowleyana*	-	+	-	+	+	+	+	+	+	+	+	+
	*S. splendens*	-	+	-	+	+	+	+	+	+	+	-	+
	*S. officinalis*	-	+	-	+	+	+	+	+	+	+	+	+
	*S. bulleyana*	-	+	-	+	+	+	+	+	+	+	+	+
	*S. digitaloides*	-	+	+	+	+	+	+	+	+	+	+	+
	*S. japonica*	-	+	-	+	+	+	+	+	+	+	+	+
	*S. plebeia*	-	+	-	+	+	+	+	+	+	+	+	+
	*S. przewalskii*	-	+	-	+	+	+	+	+	+	+	+	+
	*S. yunnanensis*	-	+	-	+	+	+	+	+	+	+	+	+
	*S. miltiorrhiza*	-	+	-	+	+	+	+	+	+	+	+	+
	*S. daiguii*	-	+	+	+	+	+	+	+	+	+	+	+
	*S. miltiorrhiza* f.*alba*	-	+	-	+	+	+	+	+	+	+	+	+
	*S. meiliensis*	-	+	-	+	+	+	+	+	+	+	+	+
	*S. hispanica*	-	+	-	-	+	+	+	+	+	+	+	+
	*S. merjamie*	-	+	+	+	+	+	+	+	+	+	+	+
	*S. sclarea*	-	+	-	+	+	+	+	+	+	+	+	+
	*S. petrophila*	-	+	-	+	+	+	+	+	+	+	+	+
	*S. tiliifolia*	-	+	-	-	+	+	+	+	+	+	+	+
	*S. chanryoenica*	-	+	+	-	+	+	+	+	+	+	+	+
	*S. yangii*	-	+	+	+	+	+	+	+	+	+	+	+
	*S. Prattii Hemsl.*	-	+	-	+	+	+	+	+	+	+	+	+
	*S. roborowskii*	-	+	-	+	+	+	+	+	+	+	+	+
	*S. nilotica*	-	+	+	+	+	+	+	+	+	+	+	+
*Rosmarinus*	*R. officinalis*	-	+	-	+	+	-	+	+	+	+	+	+
*Agastache*	*A. rugosa*	-	+	+	+	+	+	+	+	+	+	+	+
*Dracocephalum*	*D. heterophyllum*	+	+	+	+	+	+	+	+	-	+	+	+
	*D. taliense*	-	+	+	+	+	+	+	+	+	+	+	+
	*D. tanguticum*	-	+	+	+	+	+	+	+	+	+	+	+
	*D. moldavica*	-	+	+	+	+	+	-	+	+	+	+	+
*Ajuga*	*A. forrestii*	-	+	-	-	+	+	+	+	+	+	+	+
	*A. campylanthoides*	-	-	-	+	+	+	+	+	+	+	+	+
	*A. ciliata*	-	-	-	+	+	+	+	+	+	+	+	+
	*A. decumbens*	-	-	-	+	+	+	+	+	+	+	+	+
	*A. lupulina*	-	-	-	+	+	+	+	+	+	+	+	+
	*A. nipponensis*	-	-	-	+	+	+	+	+	+	+	+	+
*Leonurus*	*L. japonicus*	-	+	+	+	+	+	+	+	+	-	+	+
*Elsholtzia*	*E. densa*	-	+	-	-	+	+	+	-	+	+	+	+
*Caryopteris*	*C. trichosphaera*	-	+	-	+	-	+	+	+	+	+	+	+
	*C. mongholica*	-	+	-	+	+	+	+	+	+	+	+	-
	*C. incana*	-	+	-	+	+	+	+	+	+	+	+	+
	*C. forrestii*	-	+	-	+	+	+	+	+	+	+	+	+

The +/- refers to the presence/absence of a gene in each species that does not have the gene. ″+″: presence; ″-″ absence; * *rps*19 is across the area of LSC and IRb (add family and order information).

**Table 4 ijms-23-12080-t004:** Primers for amplifying DNA barcodes to distinguish *Salvia* species in the Lamiaceae family.

No	Species	Conserved Sequences for Designing Forward Primers	Conserved Sequences for Designing Reverse Primers	IGS
M1	*S. bowleyana*, *S. splendens*, *S. officinalis*	GCGGATATGGTCGAATGGTAAA	GCAGTTTGGTAGCTCGCAAG	*trn*G-GCC-*trn*M-CAU
M2		TGAAGTTGTCGGAATTATTTGCA	AATGCTACGCCTTGAACCAC	*ycf*3-*trn*S-GGA
M3	23 *Salvia* species	TTTTCCCCTTCCTACCCC	AAAAAAAGATGTTGCGGAGACAGGATTTGAACCCGTGACCTCAAGGTTATGAGCCTTGCGAGCTACCAAACTGCTCTACCCCGCGCTGAAGAGAAGAA	*trn*M-CAU-*atp*E
M4		TTACATAGTTATGGTTCATTTACATTAACATCTAATTAAAT	TTTTTTCATTGTACAACGAAC	*ccs*A-*ndh*D

## Data Availability

The chloroplast genome sequence data of *S. bowleyana*, *S. splendens*, and *S. officinalis* are openly available in the GenBank database with accession numbers OM617845, OM617847, and OM617846 (https://www.ncbi.nlm.nih.gov). The associated BioProject, SRA, and Bio-Sample numbers are PRJNA726222, PRJNA769231, and PRJNA769230; SAMN18926173, SAMN22106482, and SAMN22106467; SRR14415377, SRR17843445, and SRR17853381, respectively.

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
