# Peer review of "Comparative Genomics and Phylogenetic Analysis of the Chloroplast Genomes in Three Medicinal Salvia Species for Bioexploration"

_ijms, 2022, doi:10.3390/ijms232012080_

Round 1

Reviewer 1 Report

This study provides examination of the comparative genomics and phylogenetic analysis of the chloroplast genomes in three medicinal Salvia species for bioexploration of the molecular markers. Before recommending this article for publication, there are some shortcomings for that should be resolve.

Overall, the study is well designed and presented in a good way, but mostly the literature is not cited.  

Avoid starting sentences with numbers like 29, 91.

Line 33-34 should be cited.

In first paragraph discuss importance of the Lamiaceae generally and specifically of the Salvia. The following study could be cited. DOI: http://dx.doi.org/10.30848/PJB2022-3(19),   

Write full form of the abbreviations at first use like LSC and SSC regions.

Section 2.1 provide figures of different parts of the plants discussed and studied in the results.

Section 4.2. could be cited with https://doi.org/10.2298/GENSR2002435A,  

Conclusion is well justified

Author Response

Reviewer 1:

  1. This study provides examination of the comparative genomics and phylogenetic analysis of the chloroplast genomes in three medicinal Salvia species for bioexploration of the molecular markers. Before recommending this article for publication, there are some shortcomings for that should be resolve.Overall, the study is well designed and presented in a good way, but mostly the literature is not cited.  

Response: Thanks for your comments, we have cited the three more related literatures.

  1. Avoid starting sentences with numbers like 29, 91.

Response: Thanks for your advice, we have revised these sentences.

  1. Line 33-34 should be cited.

Response: Thanks for your advice, we have cited a reference.

  1. In first paragraph discuss importance of the Lamiaceae generally and specifically of the Salvia. The following study could be cited. DOI: http://dx.doi.org/10.30848/PJB2022-3(19),   

Response: Thanks for your advice, we have cited the reference as DOI: http://dx.doi.org/10.30848/PJB2022-3 (19).

  1. Write full form of the abbreviations at first use like LSC and SSC regions.

Response: Thanks for your advice,we have written the full form of LSC and SSC regions at the first use.

  1. Section 2.1 provide figures of different parts of the plants discussed and studied in the results.

Response: Thanks for your advice, we have described the plant parts that have different characteristics from the three species: Salvia bowleyana, S. splendens, and S. officinalis in the results and highlight these differences in figure 1.

  1. Section 4.2. could be cited with https://doi.org/10.2298/GENSR2002435A.

Response: Thanks for your advice, we have cited the above reference.

  1. Conclusion is well justified

Response: Thanks for your positive comments.

Reviewer 2 Report

This study by Dr. Liu and colleagues reported a nice comparative genomics analysis in three Salvia species from Lamiaceae family by using chloroplast genome sequence. Using mitochondria and chloroplast genome data to trace the phylogenetic relationship between species are very interesting. This study was designed carefully, and data representation was good. The results and conclusions of the paper were interesting. However, I found some concerns that need to be addressed before consideration for publication.

  1. What was the rationale behind this study design? Several chloroplast genomes from Salvia species were reported previously. How addition of these three genomes did add to our previous understanding? Clearly mention the manuscript, it’s largely missing.
  2. Methods have been described with details, but I think the details of the genome sequencing were missing. Authors should add more details to it.
  3. The authors may consider adding a schematic representation of the workflow of the DNA isolation, sequencing protocol and its analysis.
  4. Were these three cryptic species? What is the region behind selecting these species?
  5. I am wondering if the authors analyzed the primary structures of chloroplast encoded proteins, did they run a comparison study within Lamiaceae family?
  6. Figure 2 and 3 were not clear. 

Author Response

Reviewer 2:

This study by Dr. Liu and colleagues reported a nice comparative genomics analysis in three Salvia species from Lamiaceae family by using chloroplast genome sequence. Using mitochondria and chloroplast genome data to trace the phylogenetic relationship between species are very interesting. This study was designed carefully, and data representation was good. The results and conclusions of the paper were interesting. However, I found some concerns that need to be addressed before consideration for publication. 

  1. What was the rationale behind this study design? Several chloroplast genomes from Salvia species were reported previously. How addition of these three genomes did add to our previous understanding? Clearly mention the manuscript, it’s largely missing.

Response: Thanks for your advice, these three species have not been studies before. They taxonomic classification has not been synchronously examined using chloroplast genomes.

  1. Methods have been described with details, but I think the details of the genome sequencing were missing. Authors should add more details to it.

Response: Thanks for your advice, we have added the method of  genome sequencing in detail. 

  1. The authors may consider adding a schematic representation of the workflow of the DNA isolation, sequencing protocol and its analysis.

Response: Thanks for your advice, the workflow of DNA isolation, sequencing protocol and its analysis is quite standard. Therefore we have interpreted their procedures in the diverse parts of methods in detail (Section 4.2 and 4.3), meanwhile, we cited the two related references (No. 61 and 63) . 

  1. Were these three cryptic species? What is the region behind selecting these species?

Response: Thanks for your advice, the three species are not cryptic. We have sequenced and selected them to clearly determine the systematic plant taxonomy and relationship of the three studied species within the evolutionary course of Salvia genus in this study.

  1. I am wondering if the authors analyzed the primary structures of chloroplast encoded proteins, did they run a comparison study within Lamiaceaefamily? 

Response: Thanks for your question. Following your comments, we ran a comparison study about the primary structures of chloroplast encoded proteins from 37 Lamiaceae species in this study. The results were shown in Figure S5. And the method and results were descried in Sections 4.8 and 2.10.

  1. Figure 2 and 3 were not clear. 

Response: Thanks for your advice, we have provided the high definition figures 2 and 3 in the attached files of figures.

Round 2

Reviewer 2 Report

The authors have done a tremendous job to incorporate all the suggestions. A minor correction is required- In the "supplementary figure" section there are two "Figure S5" figures. The authors need to delete the wrong one. I recommend  to accept this article.